# Plasmonic Detection of Glucose in Serum Based on Biocatalytic Shape-Altering of Gold Nanostars

**DOI:** 10.3390/bios9030083

**Published:** 2019-06-29

**Authors:** Masauso Moses Phiri, Danielle Wingrove Mulder, Barend Christiaan Vorster

**Affiliations:** Centre for Human Metabolomics, North-West University, Potchefstroom 2520, South Africa

**Keywords:** glucose biosensor, gold nanostars, colorimetric detection, glucose oxidase, localized surface plasmon resonance, biocatalytic shape-altering

## Abstract

Nanoparticles have been used as signal transducers for optical readouts in biosensors. Optical approaches are cost-effective with easy readout formats for clinical diagnosis. We present a glucose biosensor based on the biocatalytic shape-altering of gold nanostars via silver deposition. Improved sensitivity was observed due to the nanostars clustering after being functionalised with glucose oxidase (GOx). The biosensor quantified glucose in the serum samples with a 1:1000 dilution factor, and colorimetrically distinguished between the concentrations. The assay demonstrated good specificity and sensitivity. The fabricated glucose biosensor is a rapid kinetic assay using a basic entry level laboratory spectrophotometric microplate reader. Such a biosensor could be very useful in resource-constrained regions without state-of-the-art laboratory equipment. Furthermore, naked eye detection of glucose makes this a suitable biosensor for technology transfer to other point-of-care devices.

## 1. Introduction

Great interest was sparked in the further development of improved enzyme-based biosensors for glucose monitoring [1] after Clark and Lyons introduced their glucose monitor using glucose oxidase (GOx) and an oxygen electrode [2]. The measurement of glucose is of major importance as the incidence of diabetes continues to increase due to improved global living standards [3]. Many methods have since been designed for glucose detection, including electrochemistry [1,4,5], fluorescence [6,7], surface-enhanced Raman scattering (SERS) [8,9], and surface plasmon resonance [10,11,12,13].

Electrochemical glucose biosensors are the most widely used, clinically [8]. This approach, however, suffers from a number of disadvantages such as complexity of electrode preparation, lack of stability in signal acquisition, inactivation of electrodes by the generated H_2_O_2_, relatively high cost, and reproducibility concerns [4,8,11,14,15]. Other detection strategies such as, SERS, have also been explored owing to its high sensitivity. The method has the ability to provide clear fingerprint information to identify the structures of the molecules [8,9,16,17,18]. The disadvantage of using this method for glucose measurement is that glucose has an inherent weak Raman activity making it difficult to trace it directly by Raman spectroscopy. Despite efforts for SERS enhancements, the weak surface adsorption ability of glucose produces relatively low SERS signal [8].

Optical detection approaches have the advantage of being cost-effective with an easy readout format [10,19]. A number of optical approaches for glucose sensing based on nanoparticle plasmon resonance have been reported in the past decade [10,20]. These glucose sensors are either enzyme- or non-enzyme-based assays [20,21,22]. Using enzymes has the advantage of increasing specificity of the assays [11]. GOx, isolated and produced from the fungus *Aspergillus niger,* is the most popularly used enzyme for glucose monitoring due to its very high substrate specificity [11]. It oxidizes glucose in the presence of molecular oxygen producing hydrogen peroxide (H_2_O_2_), which is used to determine glucose concentration [1,11,20].

Four main strategies have been applied in optical signal generation using nanoparticles. These include, (i) nanoparticle aggregation; (ii) surface etching; (iii) fluorescence quenching; and (iv) nanocrystal growth [10,23,24,25,26]. Nanoparticle aggregation is the most commonly employed strategy for sensing. The disadvantage with this method however, is that it lacks specificity in signal generation, as many other factors in solution may cause particle aggregation [10,11,12]. Biocatalytic growth of nanoparticles for signal amplification has been applied for the development of many assays [23,26,27,28,29]. This mechanism has allowed the tuning of the plasmonic nanoparticle shape and size resulting in different optical properties [30]. Nanostructure shape-altering mechanism of detection is one of the strategies for enhancing the sensitivity of plasmonic nanosensors. As one of the anisotropic nanocrystals, gold nanostars (AuNSs), exhibit higher refractive index sensitivity compared to spherical nanoparticles [27]. Localised surface plasmon resonance (LSPR) sensing based on shape alterations induced by an external stimulus is highly sensitive to changes in the conditions within the colloidal or detection solution [31]. Particle size growth is more sensitive with small sized nanoparticles, as they have higher absorption rates compared to larger particles like AuNSs [27].

Most nanoparticle-based glucose biosensors developed were conducted in either buffer systems, urine, or saliva [4,10,27,32,33,34,35]. Yet, venous plasma or serum are the recommended body fluids for clinical glucose diagnosis [36,37,38,39]. Some methods have however reported using biological samples [40,41]. The complexity of a biological matrix substantially increases the probability of undesired interfering and side reactions. Therefore, the use of buffered systems is likely to make false assumptions about the usability of nanomaterials in clinical diagnostics [12,41,42,43]. Plasma or serum has many proteins and lipoproteins that form a corona around the nanoparticle and changes its physiochemical properties based on the biomaterials coated around it [40]. This corona affects the effective diameter of the nanoparticles, and increases the chances of aggregation. The biomolecular corona effectively changes the synthetic identity of the nanoparticles to biological identity based on the molecules attached. Additionally, Au and Ag disintegrate in serum, making the detection in biological samples difficult [34,41,42,43,44,45,46,47,48,49,50,51,52]. There is thus, a need to develop and optimise an optical biosensor that is stable, sensitive, and robust enough for detection of clinical samples.

In this work, a sensitive, specific, and rapid optical glucose sensor based on biocatalytic shape-altering of gold nanostars AuNSs is presented. The biosensor was fabricated by optimally functionalising seedless AuNSs with GOx for enhanced stability and functionality. The functionalised AuNSs were tested for stability in various fluids and thereafter optimised for glucose sensing in spiked serum samples. Lastly, a number of analytical parameters such as specificity, kinetics, and recovery of the glucose assay were investigated.

## 2. Materials and Methods

### 2.1. Materials

Hydrochloroauric acid (HAuCl_4_), glucose oxidase (GOx), trisodium citrate, silver nitrate (AgNO_3_), ascorbic acid, sodium chloride (NaCl), polyvinylpyrrolidone (PVP) (molecular weight 10,000), hydrochloric acid (HCl), glucose, 2-(*N*-morpholino)ethanesulfonic acid (MES) at pH 6, *N*-(3-Dimethylaminopropyl)-*N*′-ethylcarbodiimide hydrochloride (EDC), sulfo-*N*-Hydroxysuccinimide (sulfo-NHS), glucose, cysteine (Cys), and phosphate buffered saline (PBS) at pH 7.4 were all purchased from Sigma-Aldrich, South Africa. Ham’s F-12K (Kaighn’s) cell culture medium was used and supplemented with 10% foetal bovine serum (FBS), which were purchased from ThermoFisher Scientific. Medidrug Basis-line S human blank serum was purchased from Industrial Analytical, South Africa. All glassware was stripped with aqua regia prior to use for synthesis. Ultrapure water (ddH_2_O) was pre-prepared with a Milli-Q ultra-pure system (18.2 MΩ/cm).

### 2.2. Preparation of AuNSs–Cys–GOx Bioconjugates

Synthesis of the PVP-stabilised seedless AuNSs and subsequent GOx bioconjugation were done using recently published methods by Phiri et al. [53,54]. Briefly, 10 mL of ddH_2_O was acidified with 10 µL of 1 M HCl followed by the addition of 50 µL of 100 mM ascorbic acid under mild stirring. Shortly after the addition of 50 µL of 50 mM HAuCl_4_ to the mixture, 50 µL of 10 mM AgNO_3_ was rapidly added to the solution which resulted in a deep blue colour change within a few seconds. Finally, 500 µL of 2.5 mM PVP was added to the mixture. Immobilisation of the enzyme onto the gold nanostars was done by adding 100 µL of 0.02 mM Cys to 2 mL of PVP-stabilised AuNSs after their clean-up, and left to incubate on a rotator for 3 h. The chemical modification of the enzyme was prepared by adding 250 mM of freshly prepared EDC/sulfo-NHS to 1 mL of GOx (5 mg/mL) in MES buffer (10 mM, pH 6) and allowed to react for 2 h. Finally, the conjugation of the AuNSs–Cys–GOx bioconjugates was accomplished by pipetting 500 µL of EDC/sulfo-NHS-modified enzymes and adding it to 2 mL of AuNSs–Cys, and incubating it overnight in the fridge. The mixture was thereafter centrifuged at 3000× *g* for 30 min to remove any unbound enzymes. The AuNSs–Cys–GOx bioconjugates were resuspended in MES buffer and stored at 4 °C until usage.

### 2.3. Characterizations and Instrumentations

UV-vis spectroscopy analyses were carried out by spectral scanning (400–990 nm) on an HT Synergy (BioTEK, VT, USA) microplate reader. The transmission electron microscopy (TEM) analyses were performed on a Tecnai F20 high-resolution transmission electron microscope (HR-TEM) at an accelerating voltage of 200 kV. Samples for TEM were prepared by applying 20 µL of nanoparticle suspension onto carbon 200 mesh copper grids (Agar Scientific, Johannesburg, South Africa), followed by drying overnight prior to imaging. ImageJ software was used to determine the particles sizes from different grids. Proton nuclear magnetic resonance (^1^H-NMR) analyses of samples in various fabrication stages were done according to the method by Venter et al. [55]. Six hundred microliters of samples were measured at 500 MHz on a Bruker Avance III HD NMR spectrometer equipped with a triple-resonance inverse (TXI) ^1^H [^15^N,^13^C] probe head and x, y, z gradient coils. ^1^H spectra were acquired as 128 transients in 32 K data points with a spectral width of 12,002 Hz. Fourier transformation and phase and base line correction were done automatically. Software used for NMR processing was Bruker Topspin (V3.5). Bruker AMIX (V3.9.14) was used for metabolite identification [56].

### 2.4. Stability of AuNSs–Cys–GOx Bioconjugates

The stability of the AuNSs–Cys–GOx bioconjugates was tested in the MES buffer, blank serum, unsupplemented- and supplemented cell culture solutions. The GOx-modified gold nanostars were centrifuged and resuspended in 200 µL of the above-mentioned fluids. UV-vis spectroscopy was used to investigate the stability of the bioconjugates and TEM analyses were done to observe the morphology and dispersity of the AuNSs in these fluids.

### 2.5. Enzyme Activity Assays

The optimised assay parameters reported in the recent study [54] were used to compare the signal generation for the determination of glucose in serum and MES buffer using AuNSs–Cys–GOx bioconjugates. The detection of glucose was assessed spectrophotometrically based on shifts in the LSPR peaks, and optically by colour changes in the solutions. Kinetic reads were done at 550 nm to determine the rate of signal generation after incubation. A number of analytical parameters were evaluated to assess the developed biosensor. Using spiked blank serum with varying concentrations of glucose, the specificity and calibration model were determined. The specificity was determined by spiking the serum with other glucose analogues such as fructose, galactose, and sucrose and their signal responses were compared under optimal conditions. Additionally, the sensing performance of the biosensor was assessed with serum lipids and cysteine instead of glucose. All experiments were carried out in triplicates.

## 3. Results and Discussion

### 3.1. Characterisation of GOx-Modified AuNSs

Figure 1A shows that UV-vis spectra of the GOx-modified AuNSs had their LSPR peak red-shift of 21.6 ± 3.2 nm from the PVP-stabilised AuNSs. The redshift for the GOx-modified AuNSs bioconjugates from the PVP-stabilised ones resulted in changes in optical properties of the AuNSs as a result of the surface functionalisation with GOx. HR-TEM images in Figure 1B shows the morphologies of the PVP-stabilised AuNSs and the enzyme-modified AuNSs. The enzyme layer attached on the peripheral surface of the AuNSs could not be imaged due to low electron resistance of protein molecules in HR-TEM examination [57]. Thus, the HR-TEM was useful for imaging the structural integrity of AuNSs after enzyme attachment. Figure 1C shows the ^1^H-NMR spectra with discernible shifts and splitting on the modified molecules. The cysteine-modified AuNSs showed that these peaks were drawn together at 3.75 ppm, indicating a shift most likely due to the specific interaction of gold with the sulphur, as observed in other studies [58,59,60]. Spectrum (II) showed the bioconjugation of cysteine-modified AuNSs with GOx (using EDC/sulfo-NHS chemistry). The peaks slightly shifted for the ester-activated enzyme covalently coupled to AuNSs–Cys, thereby indicating a successful AuNSs–Cys–GOx conjugation.

### 3.2. Stability and Characterisation of AuNSs–Cys–GOx Bioconjugates in Various Media

Figure 2A shows the stability of the GOx-modified AuNSs in different media based on change in optical densities over time. The maximum optical density of the bioconjugates in MES buffer and unsupplemented cell culture medium was observed to decline with time due to possible aggregation. However, when incubated in serum the maximum optical density increased in the period between 0–2 h and slightly declined thereafter. The increase in absorption is probably due to the complexation of unspecified serum proteins [43]. The UV-vis spectra of the bioconjugates in serum displayed better stability and only seconded by those incubated in supplemented cell culture medium. The AuNSs in the supplemented cell culture medium showed a steady decline in the UV-vis absorption in the period between 0 to 6 h before stabilising over the remaining hours. This is probably due to the FBS with proteins likely bind to nanoparticles resulting in relative better stability in the media [42,61]. Figure 2B shows the morphologies of the AuNSs–Cys–GOx bioconjugates in the different matrices. The agglomeration of the AuNSs in serum (II) is noteworthy. Agglomeration was observed for these AuNSs compared to those in other matrices. A similar observation was made for AuNSs incubated in FBS-supplemented cell culture medium (IV). The particle agglomeration observed in these two fluids could be attributed to the changes in surface properties brought about by the biomolecules and the ionic strength in the serum that forms a corona around the nanoparticles [41,43,44].

### 3.3. Optimisations of Plasmonic Glucose Detection Conditions in Serum

The foregoing observations showed that the AuNSs–Cys–GOx bioconjugates displayed both good stability in serum, as well as nanoparticle clustering. Interestingly, the nanostars clustering was recently reported to be an advantage for improving sensitivity in plasmonic assays [62]. The AuNSs–Cys–GOx bioconjugates thus—characterised by enhanced stability and improved sensitivity—were a fit candidate for biosensor fabrication as nanodevices for detection of glucose in serum. Initial attempts at signal generation using serum sample volumes ≥20 µL proved futile. Despite various attempts of detection condition optimisations, no observable colour change was observed. The failure in signal generation of the nanosensors was attributed to the nanoparticle physiochemical properties changes in complex matrices such as serum [42,44,45,61].

However, when the sample was serially diluted 1000 times, colour changes were observed as shown in Figure 3. The final concentrations of AgNO_3_ and NH_3_ in the detection solution were also optimised to a final concentration of 0.25 mM and 20 mM, respectively, to generate visible colour changes. The biosensor showed ability to differentiate between concentrations of glucose at 1000 times dilution. Sample serial dilution of 1:100–1000 was observed to be optimal for the detection of an analyte in the serum using the fabricated nanodevices. Such a great sample dilution factor offers the advantages of reducing the effects of complex sample interferences [23], and allows the quantification of samples whose concentration is very low, and whose volumes are ultra-low. For simplicity’s sake and reducing batch variations in experimentations, a dilution factor of 1:100 was chosen for further experiments that enabled the use of 2 µL of serum in a total reaction mixture of 200 µL.

### 3.4. Plasmonic Glucose Detection by Means of AuNSs Shape-Altering

Scheme 1 shows the proposed signal-generation mechanism. The silver ions, reduced by H_2_O_2_ produced from oxidised glucose, coated around the plasmonic AuNSs resulting in a shape alteration. The extent of the change in AuNSs morphology depended on the concentration of glucose in the solution. Thus, nanospheres are the potential end-result morphologies of this reaction.

The influence of the sample matrix on detection using AuNSs was investigated by detecting different concentrations of glucose measured in the MES buffer and in serum. Figure 4A shows a marked difference in the colour change depending on the sample matrix. More varied colours were observed when the measurement was done in MES buffer compared to serum. The colour changed from blue to purple to orange in MES buffer, while in serum it was from blue to intensified blue to a deep purple colour. Yet, in both matrices there was a visible distinction in colour with increasing concentrations of glucose. This provided an opportunity for this biosensor to be developed and optimised for screening of biological samples with naked eye detection.

The morphologies of AuNSs corresponding to selected solutions with different concentrations of glucose are shown in Figure 4B,C. There were clear differences in the change in morphology for AuNSs in serum and MES that explain the degree of the colour changes. In serum, the AuNSs had relatively slight changes in morphology compared to the ones in MES solution. The AuNSs in MES solution became more spherical in morphology due to the silver coating as the glucose concentration increased.

From the analysis of the nanoparticle sizes after being coated with Ag^0^ as shown in Table 1, no observable growth was found in the size compared to the changes in their morphologies. As opposed to the signal generation mechanism based on the growth of nanoparticles by addition of either Ag^+^ or HAuCl^−^ [21,23,26,28], this was observed to be merely morphology altering.

The LSPR peak shifts of AuNSs in serum in Figure 5A showed a total of 80 nm blueshift. In MES solution however, AuNSs showed a clearer and greater shift of 131 nm from the control to the highest glucose concentration as shown in Figure 5B. The reason for this observation in the two sample matrices could be due to the interferences caused by proteins and lipoproteins in the serum, which can affect the analytical performance of the biosensor [40]. When the LSPR peak shifts were plotted against the increasing concentration of glucose as shown in Figure 5A,B inserts, the correlation coefficient (*R*^2^) in both serum and MES were 0.99. This represented a predictable detectable range for glucose concentration with a 100 times dilution of the sample.

Figure 6A shows the kinetic reaction that was monitored at 550 nm from the start of incubation with all the optimal reaction conditions with different glucose concentrations. As the reaction proceeded, there was a distinct differentiation between the concentrations of glucose based on the absorbance. The biosensor only required an incubation time of <15 min at 37 °C for sufficient oxidation of glucose and to generate distinguishable colours and plasmonic shifts for detection. Figure 6B shows signal generation by addition of detection solution after incubation of the reaction mixture at 37 °C for 45 min. The biosensor was able to generate distinguishable colorimetric and plasmonic signals between the glucose concentrations in serum in less than 10 s of detection solution addition. Within 5 min, the signal generation process was near complete. This demonstrates the rapidity of the biosensor both in incubation time and detection process.

### 3.5. Analytical Performance of the Glucose Biosensor

The specificity of the glucose biosensor was investigated by using other saccharides as substrates instead of glucose. Figure 7 displays the signal response of the biosensor to these glucose analogues. No significant colour changes or LSPR peak shifts were observed in the presence of other saccharides. The biosensor however, generated a significant response when glucose was the analyte. This observation demonstrated that the signal generation was strongly dependent on the presence of glucose in the reaction solution, and not any other saccharides. The glucose biosensor demonstrated the ability to distinguish the presence and absence of glucose within the limits of detections in the presence of other structural analogues. Furthermore, when L-cysteine and lipids were used as substrates, as shown in Figure 8A,B, minimal LSPR peak shifts were observed for these analytes except for glucose. Figure 8C shows the colour changes—or lack thereof—of the different analytes. Analytes other than glucose did not produce any change in colour that was significantly different from the blank AuNSs. The biosensor thus demonstrated a specific response for glucose in the presence of potential interferences in serum.

The practical application of the fabricated biosensor was verified by investigating the recovery rates of the biosensor by determining different concentration of glucose as listed in Table 2. The results showed recovery rates of 97% to 102% for the three concentrations measured in triplicates. These recovery results were high and could meet the needs of actual clinical sample detection and quantification.

## 4. Conclusions

Here presented is a glucose biosensor based on a simple seedless synthesis of gold nanostars, functionalised in a facile way with glucose oxidase for optimal functionality. The assay used AuNSs for greater sensitivity in LSPR peak shifts and colorimetric readouts via biocatalytic altering of their morphologies. Stability in serum and sensitivity in detection was enhanced by nanostar clustering after functionalising with GOx, as well as by the shape-altering mechanism of detection. Furthermore, the sample matrix was observed to influence the colorimetric readout of the assay, with MES buffer solution being more pronounced for the naked eye detection. The biosensor was able to quantify glucose in the serum diluted 1000 times with the ability to distinguish between different concentrations. Such sensitivity can potentially be applied for measuring samples with volumes such as dried blood spots. The assay demonstrated good specificity in glucose detection. Thus, the fabricated glucose biosensor proved to be a rapid kinetic colorimetric assay that utilises a basic entry level laboratory spectrophotometric microplate reader. Such a biosensor could be very useful in resource-constrained regions of the world without state-of-the-art laboratory equipment. This biosensor is a great candidate for potential clinical diagnosis, research and development applications.

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
