# Peer review of "Plasmonic Detection of Glucose in Serum Based on Biocatalytic Shape-Altering of Gold Nanostars"

_biosensors, 2019, doi:10.3390/bios9030083_

Reviewer 1 Report

Plasmonic Detection of Glucose in Serum Based on Biocatalytic Shape-Altering of Gold Nanostars  

Masauso Moses Phiri 1*, Danielle Wingrove Mulder 1 and Barend Christiaan Vorster

The manuscript has demonstrated the use of gold nanostars for the detection of glucose in serum. The work is systematic and logical. There are some points that need to be taken into consideration. If these points are addressed, the manuscript is fit for publication.

Line 142: what does the error bar indicate for such a measurement?

Line 156: what was the concentration of GOx? If concentration of GOx is varied does the absorption peak shift or remains independent to concentration?

Line 162: Histogram needs error bar, to make the data relevant.

Line 195: the description in the figure should be more precise: What control sample mean: is this blank serum + Au NSs stabilized in PVP? Also what is detection solution? Is this the serum?

It is often that readers browse through papers by examining the figure and so figures and their descriptions are vital.

Line 216: Please sentence construction.

Line 225: Can this incubation time be decreased, by slight altering the process variables like stirring or increasing the temperature without damaging GOx?

Line 240: Error bars for inset will make the data more credible.

Line 261: Can this deposition of Ag be reversed so that the Au NSs can be reused?

Author Response

The manuscript has demonstrated the use of gold nanostars for the detection of glucose in serum. The work is systematic and logical. There are some points that need to be taken into consideration. If these points are addressed, the manuscript is fit for publication.

Point 1. Line 142: what does the error bar indicate for such a measurement?

The measurement was based in one of the UV-vis spectral reads. When all the measurements of the functionalization data were reanalyzed, the redshift was determined to be 21.6 ± 3.21 nm. The text in the manuscript has been adjusted to include this detail.

Point 2. Line 156: what was the concentration of GOx? If concentration of GOx is varied does the absorption peak shift or remains independent to concentration?

The concentration of GOx was 0.0417 mmol/L. Yes, the absorption peak shifts as the concentration of GOx varies. This has been observed in our lab and in other reported works such as (Rodríguez-Lorenzo et al., 2018). Our concentration of glucose oxidase was optimally chosen for the purpose of enabling analyte detection in first order kinetics.

Point 3. Line 162: Histogram needs error bar, to make the data relevant.

This has been done. The Figure has since been readjusted.

Point 4. Line 195: the description in the figure should be more precise: What control sample mean: is this blank serum + Au NSs stabilized in PVP? Also what is detection solution? Is this the serum? It is often that readers browse through papers by examining the figure and so figures and their descriptions are vital.

The comment is appreciated. The caption on the figure has been edited to add the extra vital details. The Control sample was blank serum with GOx-modified AuNSs, the detection solution is a signal enhancement solution comprising final concentrations of 0.25 mmol/L of AgNO3 and 20 mmol/L of NH3 in a 200 µL reaction solution.

Point 5. Line 216: Please sentence construction.

Noted and re-worked.

Point 6. Line 225: Can this incubation time be decreased, by slight altering the process variables like stirring or increasing the temperature without damaging GOx?

Yes, the incubation time can be decreased. This was observed using an assay set-up with all necessary components and running a kinetic read. When the assay was incubated at 40ºC —an optimal temperature for GOx, within 15 minutes the reaction glucose oxidation was near complete and sufficient for the signal generation that was able to distinguish varies concentrations of glucose oxidase.

Point 7. Line 240: Error bars for inset will make the data more credible.

Noted with thanks. This has been done and manuscript adjusted.

Point 8. Line 261: Can this deposition of Ag be reversed so that the Au NSs can be reused?

This cannot be done as the Ag deposits on the core of the AuNSs enabling the tuning of the shape. This is supported by some earlier EDS experimental data we carried out in our lab which showed the AuNSs containing about 12% Ag and the rest being Au. This is in agreement with a recent publication by (Atta et al., 2019) on understanding the role of AgNO3 concentrations in achieving tuneable shape control of AuNSs.

Bibliography

Atta, S., Beetz, M. & Fabris, L.  2019.  Understanding the role of AgNO 3 concentration and seed morphology in the achievement of tunable shape control in gold nanostars.  Nanoscale, 11(6):2946-2958.

Rodríguez-Lorenzo, L., de La Rica, R., Álvarez-Puebla, R.A., Liz-Marzán, L.M. & Stevens, M.M.  2018.  Addendum: Plasmonic nanosensors with inverse sensitivity by means of enzyme-guided crystal growth.  Nature materials, 17(2):205.

Reviewer 2 Report

In this manuscript, the authors demonstrated the detection of glucose in serum and MES buffer using shape-altering mechanism utilizing hydrogen peroxide (H2O2) as the enzymatic product of glucose oxidase to presence of glucose, and silver ions. The blue-shift LSPR bands of gold nanostars (AuNSs) was observed and determined as the corresponding variable to determine the concentration of glucose in the sample. Several problems are pointed out in this report on this manuscript.

1.     This manuscript contains similar images, especially for the characteristic of AuNSs and AuNS-Cys-GOx, with authors’ previous publication (Reference no. 54).

2.     The mechanism has been discussed in the author’s publication (Reference no.53). Furthermore in 2012, a previous publication (reference no. 23) by Rodríguez-Lorenzo, L., et al. which was published in Nature Materials, utilizing silver growth on AuNSs by GOx and glucose for PSA detection. However, there is different stressing in the signal transducing analysis. Please elaborate the novelty of this work.

3.     As previously mentioned, in reference no.23 of the manuscript, the concentration of GOx is essential for the kinetic of the silver growth on the AuNSs. Please explain why the selected concentration of AuNSs and GOx are used.

4.     There is a possibility of silver independence formation by H­­2O2. However, there is no visible AgNPs formation in this manuscript, based on the TEM image and the absorbance spectrum. Please explain how the side reaction can be controlled/ limited.

5.     Please consider analyzing the shape-altering mechanism in elemental analysis using EDX analysis. This collated to difference of the scheme 1 of this manuscript and the scheme 1 of reference no.53 (author’s previous work). In the previous work, the authors illustrate the crystal growth by the silver in island form. However, in this manuscript, the scheme illustrated a fully growth of the silver on the AuNSs. 

6.     In Fig. 7 and Fig. 8, the presence of the glucose showed the peak shift of LSPR up to 140 nm. However, the photograph showing the color change was different. In Fig. 7, there is no obvious purple color which was visible in Fig. 8C. Please explain.

7.     In comparison of the system in serum and MES buffer, there is no error bar in plotting the calibration line. Please consider doing the experiment more than one time. As then, the reliability of the system to show R2of 0.99 can be considered. 

8.     Please explain what the triplicate of the detection of glucose in spiked sample were. It is a repeated measurement on the results or three different experiments.

9.     The authors addressed undesired problem by the complexity of biological sample in nanomaterial-based biosensor. How this system can overcome this issue for better glucose biosensor. Please elaborate the results to the background of this work.

Author Response

Point 1. This manuscript contains similar images, especially for the characteristic of AuNSs and AuNS-Cys-GOx, with authors’ previous publication (Reference no. 54).

The images contained in this manuscript that are similar to those in reference no. 54 is because the particular study being reported in this submitted manuscript is a sequel to the previously published work. The same method of nanostars synthesis and enzyme functionalization as reported in the previous publications (Ref. # 53 and 54) were followed in this study as well. Therefore, the figures have been rightly cited and permission from co-authors was granted to be re-used in this manuscript.

Point 2. The mechanism has been discussed in the author’s publication (Reference no.53). Furthermore in 2012, a previous publication (reference no. 23) by Rodríguez-Lorenzo, L., et al. which was published in Nature Materials, utilizing silver growth on AuNSs by GOx and glucose for PSA detection. However, there is different stressing in the signal transducing analysis. Please elaborate the novelty of this work.

A few things can be pointed out with regards to the novelty and contribution of this work;

The shape/morphology-altering means of signal transduction without any significant growth in size hasn’t been reported, to the best of our knowledge. Most published works reviewed and cited mainly report the enzyme-guided growth of the nanostars as a means of signal generation. The shape-altering mechanism that was optimized in this study is reported to be more sensitive in literature, as stated in the introduction of the manuscript. Thus, this work reports that aspect of the work and demonstrates its sensitivity in signal transduction in highly diluted serum samples.

The work builds on the facile functionalization of glucose oxidase method reported by (Phiri et al., 2019) to demonstrate the optimal functionality and stability of the bioconjugates in complex sample matrix such as serum.

Lastly, it makes a significant contribution in the fabrication of nanobiosensors especially as it relates to optimizing the colour and LSPR changes in ideal buffered solutions and those in real clinical samples. The goal is to have the nanostructures be as stable and sensitive in signal transduction and colour changes in serum or urine as they are in buffer solutions.

Point 3. As previously mentioned, in reference no.23 of the manuscript, the concentration of GOx is essential for the kinetic of the silver growth on the AuNSs. Please explain why the selected concentration of AuNSs and GOx are used.

Reference no. 23 reported a study where they were building plasmonic nanosensors with inverse sensitivity to be applied in an ELISA for prostate-specific antigen (PSA). In this case, the enzyme is used as a label the reports the concentration of the antigen bound. The concentration of enzymes, in this case, will vary from assay to assay. Therefore, it was necessary for them to design the nanosensor in such a way that it was sensitive enough to detect the varying concentrations of the enzyme in each assay.

In our case, we designed an enzymatic assay that is based on the measurement of the substrate and not the enzyme. The selected concentration of the GOx (0.042 mmol/L) and AuNSs had to be constant and high enough to favour a first-order kinetic reaction in the breaking down of glucose. The concentration of both the AuNSs and GOx were carefully optimized to produce a directly proportional sensitivity in the measurement of the substrate, as opposed to Steven et al., who did an inverse one. As such, the silver concentration in the reaction system was also optimized to be able to coat the nanostars with the right kinetics to produce the desired signal transduction in colour and LSPR peak changes.

Point 4. There is a possibility of silver independence formation by H2O2. However, there is no visible AgNPs formation in this manuscript, based on the TEM image and the absorbance spectrum. Please explain how the side reaction can be controlled/ limited.

This side reaction can be controlled by reducing the concentration and volume of silver added to the reaction system, as well as using the right concentration of the base solution added to catalyze the reaction. There is also a need to see the upper limit of detection of the produced H2O2 because at that point, the silver no long coats on the nanostars but forms silver nanoparticles. The reason there were no visible AgNPs on the TEM image is that these parameters were optimized during the experiments and manuscript reports the final results of the whole experiments.

Point 5. Please consider analyzing the shape-altering mechanism in elemental analysis using EDX analysis. This collated to difference of the scheme 1 of this manuscript and the scheme 1 of reference no.53 (author’s previous work). In the previous work, the authors illustrate the crystal growth by the silver in island form. However, in this manuscript, the scheme illustrated a fully growth of the silver on the AuNSs.

The suggestion has been noted and will be explored in the follow-up work. 

Point 6. In Fig. 7 and Fig. 8, the presence of the glucose showed the peak shift of LSPR up to 140 nm. However, the photograph showing the color change was different. In Fig. 7, there is no obvious purple color which was visible in Fig. 8C. Please explain.

Colour development for the nanosensors is time-dependent, among other factors. Most of the pictures were taken within 30 minutes of the addition of the detection enhancement solution. However, the time at which all the pictures were taken for most of the images was unfortunately not standardized. This explains the difference in the two colours. If the image in figure 7 was taken a little later, the colour would have been similar to that in Figure 8.

Point 7. In comparison of the system in serum and MES buffer, there is no error bar in plotting the calibration line. Please consider doing the experiment more than one time. As then, the reliability of the system to show R2of 0.99 can be considered. 

The experiment was done in triplicates and the figures have been adjusted to depict the totality of the whole data. The error bars based on the standard deviation have also been added.

Point 8. Please explain what the triplicate of the detection of glucose in spiked sample were. It is a repeated measurement on the results or three different experiments.

These were three different experiments under the same conditions.

Point 9. The authors addressed undesired problem by the complexity of biological sample in nanomaterial-based biosensor. How this system can overcome this issue for better glucose biosensor. Please elaborate the results to the background of this work.

The nanobiosensor was developed in a stepwise manner. Initially, a feasibility detection reaction was done in a buffer solution that produced positive results. However, when the analyte was spiked in serum and then used the serum sample, there was neither colour change nor changes in the LSPR peaks. Various volumes of this serum sample were used in the assay ranging from 2 – 100 µL in a 200 µL reaction solution. There was still no response in terms of signal generation for the detection of glucose. Only when the sample was highly diluted (from 100 to 1000 times) was there observable optical and colour changes from the reaction. The dilution helped to minimize the interfering proteins and lipoproteins present in serum that results in undesirable effects when using nanostructures. The sensitivity of the nanostars was a great advantage in detection using such minute sample concentration. Unfortunately, the pictures and UV-vis spectral readings of these failed experiments were not stored for further analysis.

Bibliography

Phiri, M.M., Mulder, D.W., Mason, S. & Vorster, B.C.  2019.  Facile immobilisation of glucose oxidase onto gold nanostars with enhanced binding affinity and optimal function.  Royal Society Open Science, 6(5).

Round  2

Reviewer 1 Report

The manuscript is fit for publication.

Reviewer 2 Report

This revised MS was revised according to reviewers' comments.